# Smooth Muscle Cells of Dystrophic (mdx) Mice Are More Susceptible to Hypoxia; The Protective Effect of Reducing Ca^2+^ Influx

**DOI:** 10.3390/biomedicines11020623

**Published:** 2023-02-19

**Authors:** Arkady Uryash, Alfredo Mijares, Eric Estève, Jose A. Adams, Jose R. Lopez

**Affiliations:** 1Division of Neonatology, Mount Sinai Medical Center, Miami, FL 33140, USA; 2Centro de Biofísica y Bioquímica, Instituto Venezolano de Investigaciones Científicas, Caracas 21827, Venezuela; 3PhyMedExp, University of Montpellier, CNRS, INSERM, CHRU Montpellier, 34090 Montpellier, France; 4Univ. Grenoble Alpes, CNRS, TIMC-IMAG/PRETA (UMR 5525), 38000 Grenoble, France; 5Department of Research, Mount Sinai Medical Center, Miami Beach, FL 33140, USA

**Keywords:** vascular smooth muscle 1, hypoxia 2, Duchenne muscular dystrophy 3, muscular dystrophy 4, mdx 5, intracellular calcium 6

## Abstract

Duchenne muscular dystrophy (DMD) is an inherited muscular disorder caused by mutations in the dystrophin gene. DMD patients have hypoxemic events due to sleep-disordered breathing. We reported an anomalous regulation of resting intracellular Ca^2+^ ([Ca^2+^]_i_) in vascular smooth muscle cells (VSMCs) from a mouse (mdx) model of DMD. We investigated the effect of hypoxia on [Ca^2+^]_i_ in isolated and quiescent VSMCs from C57BL/10SnJ (WT) and C57BL/10ScSn-Dmd (mdx) male mice. [Ca^2+^]_i_ was measured using Ca^2+^-selective microelectrodes under normoxic conditions (95% air, 5% CO_2_) and after hypoxia (glucose-free solution aerated with 95% N2-5% CO_2_ for 30 min). [Ca^2+^]_i_ in mdx VSMCs was significantly elevated compared to WT under normoxia. Hypoxia-induced [Ca^2+^]_i_ overload, which was significantly greater in mdx than in WT VSMCs. A low Ca^2+^ solution caused a reduction in [Ca^2+^]_i_ and prevented [Ca^2+^]_i_ overload secondary to hypoxia. Nifedipine (10 µM), a Ca^2+^ channel blocker, did not modify resting [Ca^2+^]_i_ in VSMCs but partially prevented the hypoxia-induced elevation of [Ca^2+^]_i_ in both genotypes. SAR7334 (1 µM), an antagonist of TRPC3 and TRPC6, reduced the basal and [Ca^2+^]_i_ overload caused by hypoxia. Cell viability, assessed by tetrazolium salt (3-(4,5-dimethylthiazol-2-yl)-2,5-diphenyltetrazolium bromide, was significantly reduced in mdx compared to WT VSMCs. Pretreatment with SAR7341 increases cell viability in normoxic mdx (*p* < 0.001) and during hypoxia in WT and mdx VSMCs. These results provide evidence that the lack of dystrophin makes VSMCs more susceptible to hypoxia-induced [Ca^2+^]_i_ overload, which appears to be mediated by increased Ca^2+^ entry through L-type Ca^2+^ and TRPC channels.

## 1. Introduction

Duchenne muscular dystrophy (DMD) is a fatal muscle disorder caused by a lack of dystrophin, resulting in loss of muscle mass and function, respiratory and cardiac failure, and premature death [1]. An elevation of intracellular calcium [Ca^2+^]_i_ in dystrophin-deficient muscles is a major secondary event involved in the progression of DMD. We have previously described how the absence of dystrophin caused a chronic increase in [Ca^2+^]_i_ and sodium concentration ([Na^+^]_i_) in skeletal [2], cardiac [3], and smooth muscle cells [4], as well as neurons [5]. Dysfunctional [Ca^2+^]_i_ appears to be mediated by increased leakage of ryanodine and inositol trisphosphate receptors in the sarcoplasmic-endoplasmic reticulum and an increased influx of Ca^2+^ through transient receptor potential canonical (TRPC) channels. Studies from our laboratory have shown that reduction in [Ca^2+^]_i_ by pharmacological and non-pharmacological approaches reduced [Ca^2+^]_i_ and provoked protection in dystrophic muscle cells and neurons [2,3,4]. 

Respiratory failure and cardiomyopathy are still the two most common causes of death in patients with DMD [6]. Ventilatory alterations are observed early in the course of DMD [7]. Patients with DMD experience progressive nocturnal sleep-disordered breathing (SDB), characterized by oxygen desaturation, apnea, and hypoxia during sleep [7]. These episodes of hypoxemia have been reported to be more severe and more frequent in older patients with DMD [8,9]. Patients with DMD can potentially develop SDB in all age groups [10], in young patients as a consequence of body weight gain due to physical inactivity and prolonged corticosteroid therapy, and in older patients due to respiratory muscle weakness, specifically the diaphragm [10]. Furthermore, studies have shown that acute exposure of mdx mice to hypoxia is associated with acute myocardial alteration, characterized by diastolic and systolic dysfunction, as well as metabolic acidosis [11,12].

In this study, we tested the effects of acute hypoxic stress on [Ca^2+^]_i_ in WT and mdx VSMCs. We found that mdx VSMCs are more susceptible to hypoxic stress than WT. Susceptibility appears to depend on the anomalous increase in [Ca^2+^]_i_, and the subsequent reduction in cell viability of mdx VSMCs compared to WT VSMCs. Furthermore, Ca^2+^ influx can play an essential role as a contributor to intracellular Ca^2+^ overload observed during acute hypoxia.

## 2. Materials and Methods

### 2.1. Animal Model

Male WT-type (C57BL/10SnJ) and mdx (CS7BL/10ScSn-mdx) mice of six months originally obtained from the Jackson Laboratory (Jackson Laboratory, Bar Harbor, ME), were bred in our institution’s animal housing facility at Mount Sinai Medical Center [13]. We used mdx mice (six–seven months) because normoxic ventilation is impaired if it is associated with reduced arterial PO_2_–both features observed in DMD patients [4]. The animals were housed conventionally in a temperature- and humidity-controlled facility, operating on a 12 h light: 12 h dark cycle with food and water available ad libitum. All the protocols used in the study were carried out according to the recommendations of the Guide for the Care and Use of Laboratory Animals of the National Institutes of Health and approved by the institution (IACUC Protocol #19090).

### 2.2. Vascular Smooth Muscle Cells

The WT and mdx mice were sacrificed by inhalation of carbon dioxide overdose, causing rapid unconsciousness and death [14]. VSMCs were isolated as previously described [4]. The isolated VSMCs were cultured in a humidified atmosphere (37 °C) and for eight–ten days after plating before experimentation. A VSMCs phenotype was confirmed by validating the expression of muscle myofilament proteins (α-smooth muscle actin and β-tropomyosin) using Western blotting [13]

### 2.3. Evaluation of Isolated VSMC

VSMC isolated cells from WT and mdx mice were experimentally used if they did not shorten when perfused with the Ca^2+^ containing a Ringer solution (1.8 mM Ca^2+^), and (ii) they responded to electrical stimuli (1 ms square pulse duration, ~1.5× threshold voltage).

### 2.4. Ca^2+^ Selective Microelectrodes and Resting [Ca^2+^]_i_ Measurements

Double-barreled Ca^2+^ selective microelectrodes were manufactured as previously described [15]. Briefly, microelectrodes were pulled from acid-EGTA washed glass using a modified microelectrode puller (Industrial Science type M-1); the puller was fine-tuned to produce abruptly tapered microelectrodes with submicrometer tips (tip outside diameter < 1 µm) in order that the VSMCs could be impaled without producing a local injury. The tip resistances of each barrel were 10–16 MΩ when filled with 3 M-KC1. Air bubbles were removed from the tip of the microelectrode with a tapered glass filament. The resting membrane potential (RMP) was used as an indicator of the muscle cell condition and penetration of the microelectrode membrane. VSMCs were sometimes damaged during impalement, as evidenced by membrane depolarization and the development of a local injury around the impalement site. These recordings from those cells were discarded and not included in the data analysis.

The Ca^2+^ microelectrode barrel was made hydrophobic by silanization with dimethyldichlorosilane vapor (Sigma-Aldrich, St. Louis, MO, USA) and baked at 150 °C for 1 h. Subsequently, the salinized tip of the microelectrode was filled with a liquid ion sensor ETH 129 (Fluka-Sigma-Aldrich, MO, USA), and the rest of the barrel was filled with pCa7. The barrel to record the RMP was filled with filtered 3M KC1 (Vm). Ca^2+^ selective microelectrodes were calibrated in calcium buffer solutions (pCa3-8) before and after intracellular Ca^2+^ determinations [15]. Only those that showed a linear response (29.5 mV/decade, at 37 °C) between pCa3—pCa7 were used for Ca^2+^ measurements [15]. Microelectrodes showed reproducible upper and lower detection limits in vitro calibration and did not show significant changes in the slope (between pCa3 and pCa7) after pre-incubation with nifedipine or SAR7334 at the concentrations used in this study.

As previously described, isolated smooth muscle cells from WT and mdx mice were impaled with a precalibrated Ca^2+^-selective microelectrode [4]. Resting membrane potential (Vm) and Ca^2+^ potential (V_Ca_) were recorded using a high-impedance amplifier (WPI Duo 773 amplifier, World Precision Instruments, Sarasota, FL, USA) [4,15]. The specific Ca^2+^ potential (V_CaE_) was obtained by subtracting the 3M KCl barrel potential (Vm) from the Ca^2+^ barrel potential (V_Ca_). Only impalements that met the following stringent criteria were experimentally used: *(i)* the RMP in normoxic VSMCs was not less negative than −50 mV; *(ii)* the Vm and differential signal (V_CaE_) drifted no more than 3 mV during the recording period; *(iii)* microelectrode post-calibration (after muscle withdrawal) showed a linear response between pCa 6 and 7. 

### 2.5. Acute Hypoxia Protocol

Hypoxia was induced by perfusing WT and mdx VSMCs with a glucose-free solution saturated with 95% N_2_-5% CO_2_ for 30 min (PO_2_ of 5–10 mmHg in the perfusion solution). [Ca^2+^]_i_ was measured under normoxic conditions (95% air, 5% CO_2_ conditions) and after 45 min of hypoxia.

### 2.6. Vascular Smooth Muscle Cell Viability

The viability of WT and mdx VSMCs was determined under normoxic and hypoxia conditions using the 3 (4,5-dimethylthiazol-2-yl)-2,5-diphenyltetrazolium bromide (MTT) assay according to the manufacturer’s kit instructions (Abcam, Waltham, MA, USA). Furthermore, the viability of the VSMC was determined in both genotypes pretreated with nifedipine or SR7334 and then subjected to hypoxia. Data collected from treated and untreated WT and mdx VSMCs are represented as a reduction in the MTT signal relative to untreated WT VSMCs.

### 2.7. Solutions

The Ringer solution had the following composition (in mM): 135 NaCl, 5 KCl, 1.8 CaCl_2_, 1 MgCl_2_, 5 glucose, and 3.6 NaHCO_3_ (pH 7.4). The Ringer solution was aerated with a mixture of 95% O_2_ and 5% CO_2_ under normoxic conditions. For the low Ca^2+^ solution, CaCl_2_ was omitted and instead 2 mM MgCl_2_ and 1 mM ethylene glycol tetraacetic acid (EGTA) were added, reducing [Ca^2+^]_e_ to 35 ± 8 nM as determined with Ca^2+^ selective microelectrodes. Nifedipine (10 µM) and SAR7334 (1 µM) were prepared from stock solutions.

### 2.8. Statistical Analysis

Data are presented as a mean ± standard deviation of the mean. One-way analysis of variance (ANOVA) on the ranks was used, followed by Tukey’s post hoc tests. *p* < 0.05 was considered significant. n*_mice_*: number of mice and n*_cell_*: number of measurements. A GraphPad Prism 9 (GraphPad Software, Inc.,Boston, CA, USA) was used for the statistical analysis. 

## 3. Results

### 3.1. Aberrant Resting Membrane Potential and [Ca^2+^]_i_ in Mdx Vascular Smooth Cells

In resting muscles, the steady state value of [Ca^2+^]_i_ is the net result of influxes, sequestration, and calcium extrusion. We confirm that under normoxia conditions, there was a significant difference in resting membrane potential and [Ca^2+^]_i_ in mdx VSMCs compared to WT [4]. The resting membrane potential (RMP) in WT was −63 ± 2 mV, while in mdx it was −54 ± 3 mV (*p* < 0.001 compared to WT) (Figure 1A). [Ca^2+^]_i_ in WT VSMCs was (123 ± 3 nM) while in mdx it was (285 ± 31 nM) (*p* < 0.001 compared to WT) (Figure 1B). 

### 3.2. Hypoxia-Induced Muscle Depolarization and [Ca^2+^]_i_ Overload

Acute hypoxia-induced muscle membrane depolarization and [Ca^2+^]_i_ overload in WT and mdx VSMCs, however, resulted in the magnitude being significantly greater in mdx than in WT VSMC. In WT VSMCs, hypoxia caused a consistent depolarization of the RMP from −63 ± 2 mV (normoxic) to −54 ± 3 mV, while in mdx VSMCs, it went from −54 ± 3 mV to −41 ± 3 (Figure 1A). Similarly, hypoxia-induced reproducible [Ca^2+^]_i_ overload in WT VMSCs increased from 123 ± 3 nM (normoxic) to 980 ± 119 nM and in mdx VSMCs from 285 ± 31 nM to 2962 ± 287 nM (Figure 1B). A persistent high intracellular elevation of [Ca^2+^] induced by hypoxia caused the development of hypercontracture and death in mdx VSMCs.

### 3.3. Reduction of Extracellular Ca^2+^ Prevented Hypoxia-Induced [Ca^2+^]_i_ Overload

The effect of a low (35 ± 8 nM) extracellular Ca^2+^ concentration ([Ca^2+^]_e_) on [Ca^2+^]_i_ was evaluated under normoxia and acute hypoxia in VSMCs from WT and mdx mice. We confirmed that the reduction in [Ca^2+^]_e_ caused a significant decrease in resting [Ca^2+^]_i_ in WT and mdx VSMCs and that the magnitude was greater in mdx than in the WT VMSCs [4]. In mdx VSMCs [Ca^2+^]_i_ was reduced by 55% (from 285 ± 31 nM to 128 ± 10, *p* < 0.001), while in WT, it was reduced by 23% (from 122 ± 3 nM to 94 ± 5 nM, *p* < 0.001) in VSMCs (Figure 2). In addition to the decrease in [Ca^2+^]_i_ in WT and mdx VSMCs, incubation in a low (35 ± 8 nM) Ca^2+^ solution markedly reduced the increase in [Ca^2+^]_i_ overload secondary to hypoxia in both genotypes (compared with Figure 1B); however, the inhibitory effect was more marked in mdx VSMCs compared to WT (Figure 2 insert). The reduction in [Ca^2+^]_e_ prevented the development of hypoxia-induced hypercontracture in mdx VSMCs.

### 3.4. Nifedipine Did Not Modify Resting [Ca^2+^]_i_ but Prevented Hypoxia-Induced Elevation of [Ca^2+^]_i_

To further elucidate the molecular mechanisms that result in elevation [Ca^2+^]_i_ during hypoxia, in another set of experiments, we studied the effects of nifedipine, an L-type Ca^2+^ channels blocker, on [Ca^2+^]_i_ in WT and mdx VSMCs under normoxic and hypoxic conditions. Under normoxic conditions, incubation in nifedipine 10 µM for 10 min did not modify resting [Ca^2+^]_i_ in WT or mdx VSMCs compared to untreated VSMCs (Figure 3). Nifedipine (10 µM) was used to fully block the Ca^2+^ current in VSMCs (Lopez unpublished results). However, nifedipine partially prevented the magnitude of elevation of [Ca^2+^]_i_ associated with hypoxia in both genotypes, with effects on mdx VSMCs greater than WT (Figure 3). In WT VSMCs pretreated with nifedipine, hypoxia elevated [Ca^2+^]_i_ to 760 ± 52 nM (22% less than in untreated WT, *p* < 0.001) while mdx VSMCs to 2013 ± 366 nM (32% less than in untreated mdx, *p* < 0.001) (Figure 3). Nifedipine did not modify the RMP in WT and mdx VSMCs. Nifedipine did not prevent the development of hypercontracture and cell death in mdx VSMCs during a hypoxic episode. Similar results of nifedipine on [Ca^2+^]_i_ during hypoxia were obtained when WT and mdx VSMCs were pretreated with amlodipine and other L-type Ca^2+^ channel blockers .

### 3.5. Contribution of TRPC-Mediated Ca^2+^ Entry to Hypoxia-Induced Increase in [Ca^2+^]_i_

We have shown above that acute hypoxia caused an alteration in intracellular Ca^2+^ dyshomeostasis in WT and mdx VMSCs. Therefore, we studied the effect of SAR7334, a TRPC6 and TRPC3 channel blocker [15], which reduces [Ca^2+^]_i_ in VSMCs [4]. We confirm [4] that preincubation with SAR7334 (1 µM) caused a reduction in [Ca^2+^]_i_ in WT VSMCs from 122 ± 3 nM to 100 ± 6 nM (18% reduction) and mdx VSMCs from 285 ± 31 nM to 154 ± 15 (45% reduction) (Figure 4). Furthermore, SAR7334 at 1 µM reduced the magnitude of elevation of [Ca^2+^]_i_ associated with hypoxia in both genotypes. In WT SAR7334 pretreated VSMCs, hypoxia caused an elevation to 342 ± 38 nM (760 ± 52 nM in untreated cells) and in mdx VSMCs to 837 ± 95 nM (2013 ± 366 nM in untreated cells) (Figure 4). SAR7334 decreased the hypoxia-induced elevation of [Ca^2+^]_i_ by 64% compared to untreated WT VSMCs (*p* < 0.001) and 72% in mdx VSMCs compared to untreated mdx VSMCs (*p* < 0.001) (Figure 4 insert). SAR7334 did not modify hypoxia-induced depolarization in either genotype. Pretreatment with SAR7341 partially prevented the development of hypoxia-induced hypercontracture in mdx VSMCs.

### 3.6. SAR7334 Improved VSMC Viability Subjected to Hypoxia

As shown in Figure 5, cell viability, as assessed by MTT, was significantly reduced in mdx compared to WT VSMCs. Hypoxia decreased cell viability in both genotypes, the effects being more significant in mdx than in WT VSMCs. Pretreatment with SAR7334 increased cell viability in normoxic mdx (*p* < 0.001) without any effect on WT VSMCs (*p* = 0.91). Furthermore, SAR7334 increased cell viability in both groups during hypoxia compared to the untreated WT and mdx VSMCs (*p* < 0.001, respectively).

## 4. Discussion

Although abnormalities in intracellular [Ca^2+^] in smooth muscle cells associated with hypoxia have been suggested, this is the first study to examine the effect of acute hypoxia on [Ca^2+^]_i_ in mdx VSMCs. The main findings of the present study are *(i)* hypoxia elicited depolarization of the muscle resting membrane potential and an elevation in [Ca^2+^]_i_ in both WT and mdx, whose magnitude is much higher in mdx than in WT VSMCs. *(ii)* The removal of extracellular Ca^2+^ blocked intracellular Ca^2+^ elevation caused by hypoxia. *(iii)* Pretreatment with nifedipine reduced the amplitude of hypoxia-induced increase in [Ca^2+^]_i_ in both genotypes. *(iv)* Blockade of the TRPC3 and -6 channels abolished the increase in [Ca^2+^]_i_ elicited by hypoxia. *(v)* Cell viability was significantly reduced in mdx compared to WT VSMCs. Hypoxia decreased cell viability in both genotypes, and pretreatment with SAR7334 increased cell viability in normoxic mdx and WT and mdx during hypoxia.

DMD is a progressive muscle-wasting disease caused by the absence of the protein dystrophin [1]. Dystrophin is a 427-kDa cytoskeletal protein that is the major component of the dystrophin-glycoprotein complex, which bridges the inner cytoskeleton and the extracellular matrix in muscle fibers [16]. Dystrophin expression has been identified in various tissues such as the skeletal and cardiac muscle, uterus, and brain, as well as the smooth muscle, where it plays a critical role in maintaining the integrity of the sarcolemma [17]. The lack of dystrophin in the smooth muscle has been associated with different alterations in the respiratory and gastrointestinal tract, and vascular bed [18,19,20]. 

Sleep disorder breathing with hypoxia is a dominant pathological component of the complex changes observed in patients with DMD; however, no studies have examined the role of hypoxia on [Ca^2+^]_i_ in striated and non-striated muscle cells. Therefore, we characterize the acute effects of hypoxia on [Ca^2+^]_i_ in VSMCs isolated from WT and mdx mice.

In excitable cells [Ca^2+^]_i_ is maintained at low concentrations (100–120 nM) against a large extracellular concentration gradient [5,21]. [Ca^2+^]_i_ is normally maintained by a complex equilibrium between the activities of Ca^2+^ release channels, sarco-endoplasmic reticulum-ATPase reuptake pumps, a Ca^2+^/Na^+^ exchanger, intracellular Ca^2+^ buffers, and surface membrane channels [22]. An increase in [Ca^2+^]_i_ is generally accepted to be a crucial event in dystrophic pathogenesis [23,24,25], including VSMCs [4]. Our data confirmed that quiescent mdx VSMCs have higher intracellular [Ca^2+^] compared to non-dystrophic WT VSMCs. Failure to maintain a low [Ca^2+^]_i_ in VSMCs may activate enzymes such as proteases, nucleases, and lipases that impair energy production, compromise muscle function and cause cell death [26]. A similar aberrant elevation in [Ca^2+^]_i_ has been observed in striated muscle cells from patients with DMD and mdx mice [24]. The magnitude of elevation of [Ca^2+^]_i_ in normoxic VSMCs compared to WT is quantitatively similar to those observed in skeletal muscle cells from patients with DMD [24] and dystrophic mdx muscle [2]. Furthermore, studies in our laboratory provide evidence that normalization of [Ca^2+^]_i_ in dystrophic muscle cells, including VSMCs reduced oxidative stress, calpain activity, muscle damage, and improved muscle cell viability and function [27]. Thus, the weight of evidence supports the notion that [Ca^2+^]_i_ plays a crucial role in the pathogenesis of DMD, and preventing its overload reduces muscle injury and minimizes hypoxia-induced cell damage. 

We found that hypoxia was associated with muscle membrane depolarization in WT and mdx; however, the magnitude was much greater in mdx than in WT VSMCs. Previous evidence implies that hypoxia blocks voltage-gated K^+^ channels in vascular smooth muscle cells, causing membrane depolarization [28]. Hypoxia also caused an elevation in [Ca^2+^]_i_ in both genotypes; however, the rise was more significant in mdx than in WT VSMCs. This result is consistent with an increase in sarcolemmal Ca^2+^ influx in both genotypes; however, it can also be associated with a decrease in Ca^2+^ uptake by the sarcoplasmic reticulum and/or a decrease in Ca^2+^ efflux due to inhibition of the Na^+^/Ca^2+^ exchanger. Due to these possibilities, we explored the contribution of the sarcolemmal Ca^2+^ influx to intracellular Ca^2+^ overload in WT and mdx VSMCs. Incubation in low Ca^2+^-solution prevented hypoxia-induced elevation of [Ca^2+^]_i_ in WT (86% inhibition) and mdx VSMC (93% in inhibition). Although the other Ca^2+^ pathways were not studied, it is fair to conclude that most of the Ca^2+^ involved in the intracellular overload observed during hypoxia is related to nifedipine-SAR 7334 sensitive sarcolemmal Ca^2+^ influx.

To characterize the Ca^2+^ sarcolemmal influx associated with hypoxia, WT and mdx VSMCs were treated with nifedipine. Nifedipine is an L-type Ca^2+^ channel blocker, which prevents the entry of Ca^2+^, resulting in a reduction in oxidative stress, improved muscle function, and an antiproliferative effect in VSMCs [29,30]. Nifedipine did not modify resting [Ca^2+^]_i_ in normoxic WT and mdx VSMCs, but partially prevented elevation in [Ca^2+^]_i_ caused by hypoxia (22% in WT versus 32% mdx) (see Figure 3). The effect of nifedipine could be related to its blocking effect on plasma membrane Ca^2+^ influx through L-type Ca^2+^ channels, which could be partially activated due to the membrane depolarization observed during acute hypoxia [31]. Nifedipine did not repolarize mdx VSMCs under normoxia or WT and mdx during hypoxia . The lack of a nifedipine effect on RMP in the normoxic condition is consistent with previous studies carried out in the papillary muscle [32].

To gain insight into hypoxia-induced Ca^2+^ influx, we studied the effects of SAR7334, a TRPC3 and TRPC6 blocker [33], on WT and mdx VSMCs. TRPC channels are non-selective cation channels that are permeable to monovalent and divalent cations, including Ca^2+^ and Na^+^, and represent the main pathway of Ca^2+^ entry in excitable and non-excitable cells [34]. TRPC channels are expressed in various cell types, including VSMCs, where they play an essential role in mediating Ca^2+^ cellular signaling [35,36]. Muscular dystrophy has been associated with an increased expression and activation of TRPC channels in mature muscle fibers and myotubes [25,34]. Dysregulation of TRPC channels has been associated with severe pulmonary disease reported in children and adolescents with DMD [37]. We have previously shown that SAR7334, a TRPC channels buckler [33] reduced [Ca^2+^]_i_ and prevented the influx of Ca^2+^ induced by 1-oleoyl-2-acetyl-sn-glycerol in WT and mdx VSMCs; furthermore, mdx VSMCs showed a significant up-regulation of TRPC-3, -6 proteins compared to WT VSMCs [4]. The present data corroborated our previous findings on the effect of SAR7334 on resting [Ca^2+^]_i_ in WT and mdx VSMCs [4] but also showed the blocking effect of SAR7334 on the acute hypoxia-induced elevation of [Ca^2+^]_i_ in both genotypes. 

The underlying mechanism of SAR7334 on [Ca^2+^]_i_ in both genotypes appears to be independent of the effect on the resting membrane potential, because it did not reverse the depolarization associated with hypoxia in WT and mdx VSMCs . SAR7334 was used at a high concentration (1 µM) to block both TRPC3 and 6 channels since concentrations <1 µM block preferentially TRPC6 channels but not TRPC3 [33]. Although TRPC3 and 6 may not be the only TRPC channels that mediate hypoxia-induced elevation of [Ca^2+^], it appears to account for most of the response, as shown in those VSMCs pretreated with SAR7334. These findings suggest that a Ca^2+^ influx mediated through TRPC channels plays a critical role in the elevation of [Ca^2+^]_i_ observed during acute hypoxia in WT and mdx VSMCs, a contribution that appears to be more important in mdx than in WT. Therefore, it is possible that blocking TRPC3 and 6 may have a therapeutic potential to prevent intracellular Ca^2+^ muscle overload in patients with DMD.

Intracellular [Ca^2+^] has been suggested as a vital regulator of cell survival and cell death in response to stress conditions. Chronic increases in intracellular [Ca^2+^] can induce the production of reactive oxygen species (ROS), activation of proteases as calpain, and the release of factors that promote apoptosis [38]. Intracellular Ca^2+^ and ROS signaling are closely integrated [39]. Thus, intracellular Ca^2+^ can modify mitochondrial ROS generation and vice versa [39]. Under normoxic conditions, mdx VSMCs showed lower cell viability than WT VSMCs. Chronic elevated [Ca^2+^]_i_ and subsequent increases in intracellular ROS generation may explain this low cell viability. Using the DCFDA fluorescence probe for measurements of oxidant levels, we found that mdx VSMCs had a significantly increased DCF fluorescence signal rate compared to controls (Lopez unpublished results). Hypoxia caused a reduction in cell viability in WT VSMCs and a further reduction in mdx VSMCs compared to untreated cells. Pretreatment with SAR7334 significantly increased cell viability in WT and mdx VSMCs. It is reasonable to suggest that the effects of SAR7334 could be related to a reduction in [Ca^2+^]_i,_ which prevents the activation of pro-apoptotic signaling pathways, which in turn will reduce cell viability. 

### 4.1. Study Limitations

Although we have demonstrated unequivocally in this study that mdx VSMCs are less tolerant to hypoxia than WT, some limitations should be noted. Nifedipine was used at a high concentration which could interact with other ion channels in muscle cells. The leakage of ryanodine and inositol trisphosphate receptors, as possible contributors to Ca^2 +^ overload during hypoxia, was not explored in either genotype. 

### 4.2. Clinical Consideration

Patients with DMD are prone to suffering progressive nocturnal sleep-disordered breathing and hypoxemia [40,41]. Insufficient ventilation in dystrophic patients is secondary to a weakened diaphragm, which can lead to transient periods of hypoxia. More than 70% of patients with DMD show symptoms of respiratory disease, even those who receive ideal medical treatment [42]. Respiratory and cardiac diseases are the leading causes of death in patients with DMD; in fact, 90% of dystrophic patients succumb to respiratory or cardiac failure. In the present study, we found that mdx VSMCs are more sensitive to hypoxia-induced intercellular Ca^2+^ overload than WT. This finding is important due to the frequency with which patients with DMD are subjected to episodes of hypoxia. The elevation in [Ca^2+^]_i_ induced by hypoxia may be a common mediator contributing to cell injury in VSMCs and other tissues, such as cardiac cells and neurons. In mdx cardiac cells, acute hypoxia causes diastolic and systolic dysfunction [11]. 

Furthermore, mdx neurons appear to be more susceptible to hypoxic insults than WT, a vulnerability that can contribute to the development of cognitive deficits in patients with Duchenne muscular dystrophy. Furthermore, this study opens new perspectives on the mechanisms of this devastating genetic disease and raises exciting issues of potential therapeutic relevance for the pharmacological management of such periods of hypoxia in patients with DMD. Current therapies for DMD can slow the progression of the disease but do not target abnormal intracellular Ca^2+^ handling and subsequent intracellular Ca^2+^ overload.

## 5. Conclusions

These studies demonstrate that hypoxia caused muscle membrane depolarization and elevation of [Ca^2+^]_i_ in WT and mdx VSMCs. The lack of dystrophin made mdx VSMCs more susceptible to hypoxia-induced Ca^2+^ overload than WT VSMCs. The [Ca^2+^]_i_ overload associated with the hypoxic episode appears to be mediated by two independent Ca^2+^ entry pathways, the TRPC channel (SAR7334 experiments) and the L-type Ca^2+^ channels (nifedipine experiments). However, TRPC channels appear to be the main path responsible for intracellular Ca^2+^ elevation. Mechanistically, muscle membrane depolarization does not appear to be the primary mechanism for hypoxia-induced elevation of [Ca^2+^]_i_, because nifedipine and SAR7334 ameliorate or prevent elevation of [Ca^2+^]_i_ without reversing muscle depolarization.

## Figures and Tables

**Figure 1 biomedicines-11-00623-f001:**
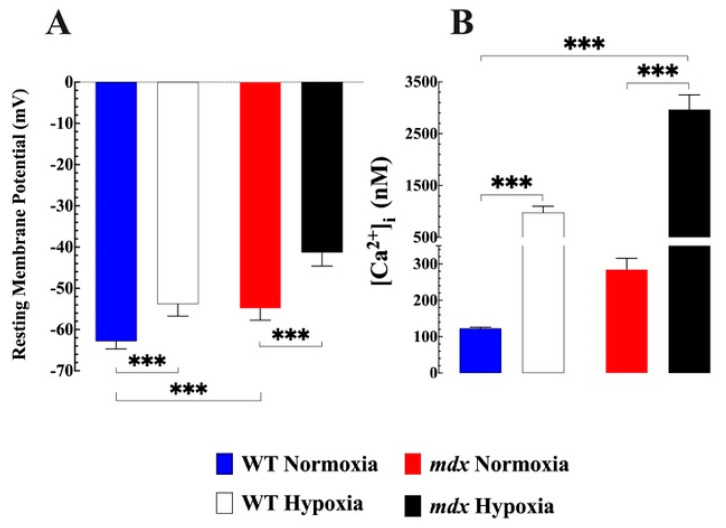
***Effects of hypoxia on resting membrane potential and [Ca^2+^]_i_ in WT and mdx VSMCs*.** (**A**) The average resting membrane potential in normoxic WT was −63 ± 2 mV, while in mdx it was 54 ± 3 mV (*p* < 0.001 compared to WT). Acute hypoxia caused a significant depolarization in WT (52 ± 3 mV) and mdx (41 ± 3 mV). (**B**) [Ca^2+^]_i_ in the WT VSMCs was (123 ± 3 nM) while in *mdx* it was significantly more elevated (285 ± 31 nM (*p* < 0.001 compared to WT). Hypoxia caused in WT VSMCs an elevation of [Ca^2+^]_i_ to 980 ± 119 nM and in mdx to 2962 ± 287 nM (*p* < 0.001 compared to WT). *n*_mice_ = 5 WT and 7 mdx, *n*_cell_ = 17–23 for WT, and 11–20 for mdx. The values were expressed as means ± S.D; *** *p* < 0.001.

**Figure 2 biomedicines-11-00623-f002:**
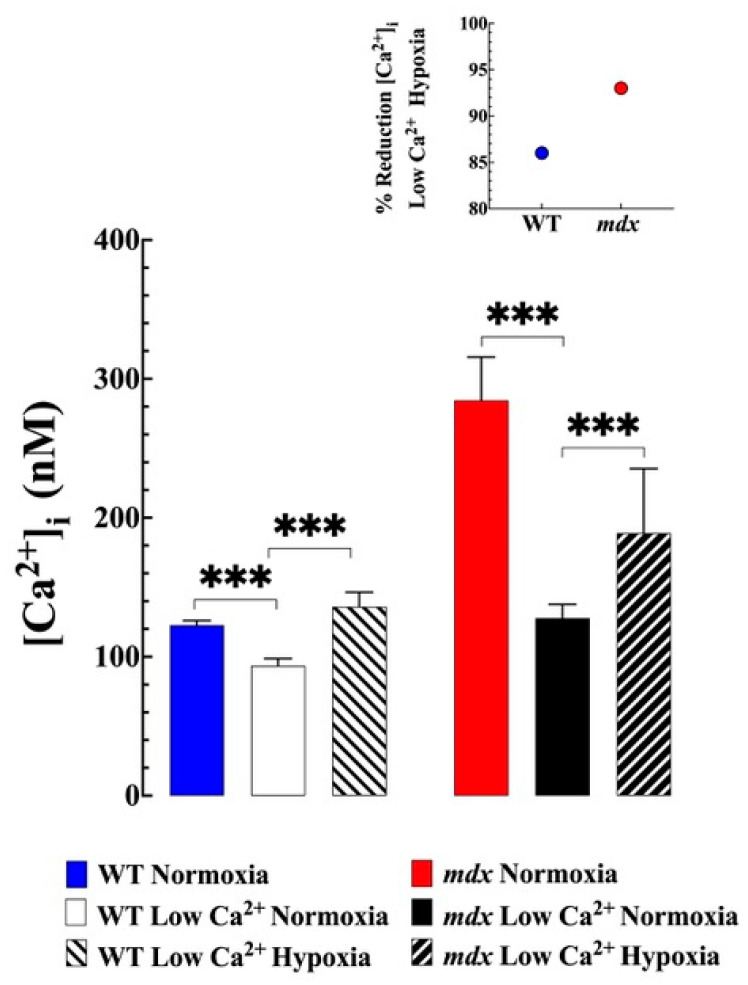
***The role of extracellular Ca^2+^ in hypoxia-induced [Ca^2+^]_i_ overload*.** Reduced [Ca^2+^]_e_ caused a significant decrease in [Ca^2+^]_i_ in WT from 122 ± 3 to 94 ± 5 nM (*p* < 0.0001 compared to untreated) and *mdx* VSMC from 285 ± 31 nM to 128 ± 10 (*p* < 0.0001 compared to untreated). Incubation in a low Ca^2+^ solution consistently inhibited hypoxia-induced increases in [Ca^2+^]_i_ in both genotypes. The insert shows the percentage of inhibition induced by lowering [Ca^2+^]_e_ in acute hypoxia. *n*_mice_ = 3 per genotype, *n*_cell_ = 17–23 for WT and 17–20 for *mdx*. The values were expressed as means ± S.D; *** *p* < 0.001.

**Figure 3 biomedicines-11-00623-f003:**
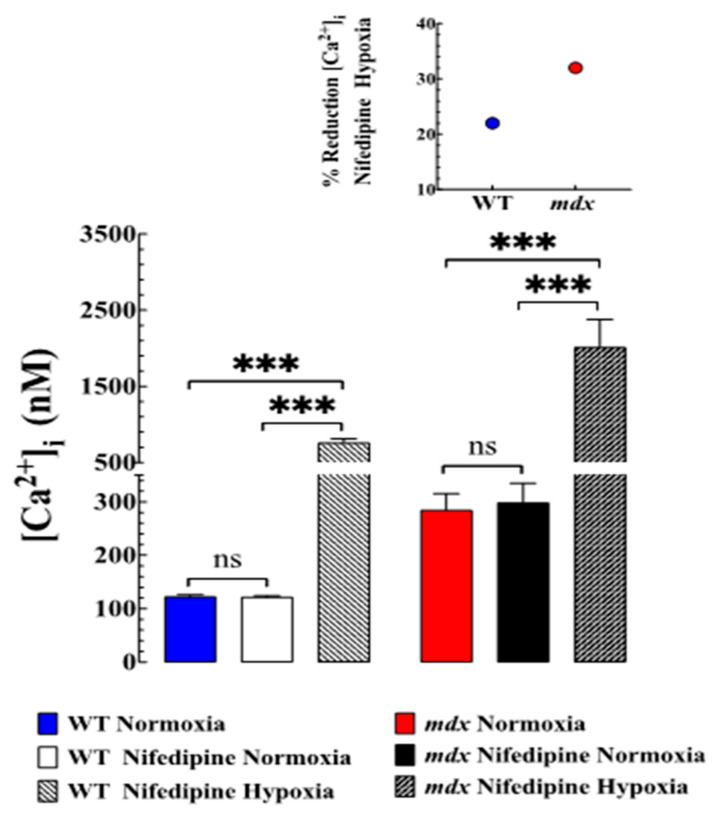
***Effects of nifedipine on**hypoxia-induced alteration of [Ca^2+^]_i_*_._** Nifedipine 10 µM did not modify resting [Ca^2+^]_i_ in WT or *mdx* VSMC compared to untreated VSMCs. However, nifedipine partially prevented the magnitude of elevation of [Ca^2+^]_i_ associated with hypoxia in both genotypes. In WT VSMCs pretreated with nifedipine [Ca^2+^]_i_ was 760 ± 52 nM (*p* < 0.0001 compared to untreated WT,) while the *mdx* VSMCs was 2013 ± 366 nM ( *p* < 0.0001compared to untreated *mdx.* The insert shows the percentage of inhibition induced by nifedipine under acute hypoxia. *N*_mice_ = 4 per genotype, *n*_cell_ = 14–23 for WT and 14–20 for *mdx*. The values were expressed as means ± S.D; ns *p* > 0.05, *** *p* < 0.001.

**Figure 4 biomedicines-11-00623-f004:**
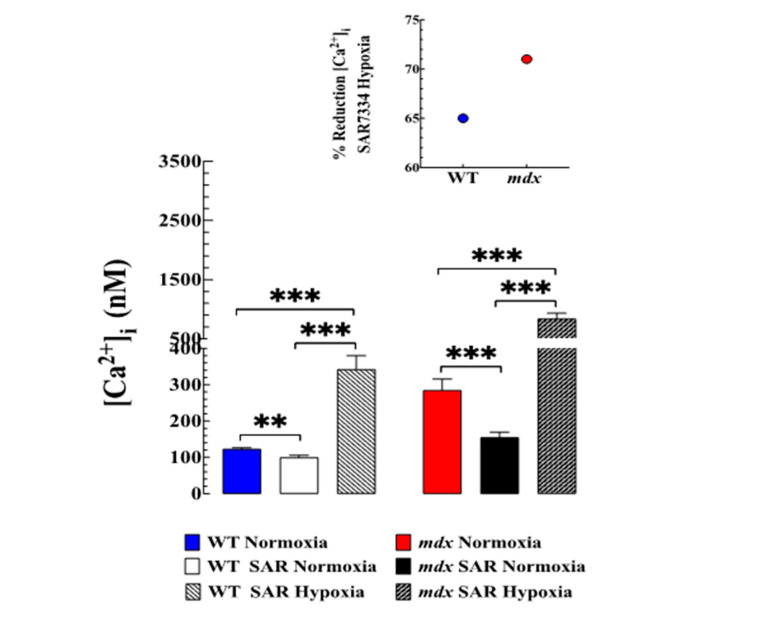
***SAR7334 blocked hypoxia-induced elevation of [Ca^2+^]_i_*.** [Ca^2+^]_i_ was measured in VSMCs isolated from WT and *mdx* mice before and after incubation in SAR7374 (1 µM), as well as during hypoxia. Preincubation in SAR7374 significantly reduced [Ca^2+^]_i_ in WT VSMCs from 123 ± 3 nM to 100 ± 6 nM and in *mdx* VSMCs from 285 ± 31 nM to 154 ± 15 nM. In WT SAR7334 pretreated VSMCs hypoxia caused an elevation to 342 ± 38nM and in *mdx* VSMCs to 837 ± 95 nM. The insert shows the percentage of inhibition induced by SAR7341 during hypoxia. *N*_mice_ = 4 per genotype, *n*_cell_ = 15–23 for WT and 15–20 for *mdx*. The values were expressed as means ± S.D; ** *p* < 0.01, *** *p* < 0.001.

**Figure 5 biomedicines-11-00623-f005:**
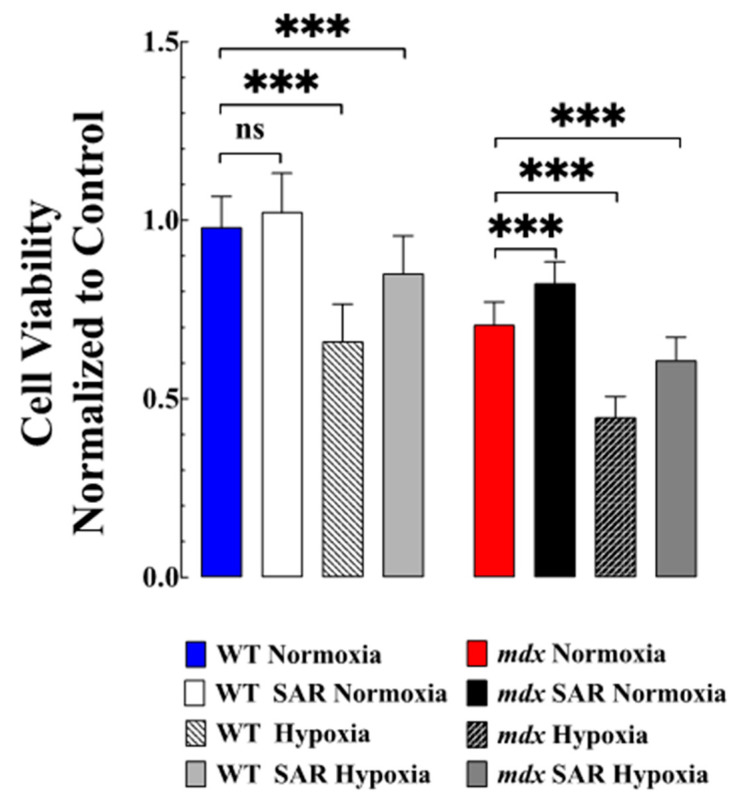
***Effects of SAR7334 on cell viability***. The viability of *mdx* VSMC was significantly less than in WT VSMC under normoxic conditions and SAR7334 treatment increased the viability of *mdx* VSMC compared to untrtreated *mdx* VSMC (*p* < 0.001). Hypoxia decreased cell viability in WT and *mdx* VSMCs and pretreatment with SAR7334 increased cell viability in both groups during hypoxia, compared to untreated WT and *mdx* VSMC (*p* < 0.001, respectively). Cell viability data was obtained from *n*_mice_ = 3/group and *n*_cells_ = 10–15/group. All values were normalized to the control untreated VSMCs and expressed as mean ± S.D; *** *p* < 0.001.

## Data Availability

Data are contained within the article or are available upon request from the corresponding author.

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
