# Peer review of "Smooth Muscle Cells of Dystrophic (mdx) Mice Are More Susceptible to Hypoxia; The Protective Effect of Reducing Ca2+ Influx"

_biomedicines, 2023, doi:10.3390/biomedicines11020623_

Round 1
Reviewer 1 Report
Hypoxia and sleep apnea are characteristic for Duchenne muscular dystrophia (DMD) patients. Uryash et al studied the effects of DMD on calcium concentration in vascular smooth muscle cells (VSMC) by comparing VSMC derived from wildtype and mdx (murine model of DMD) mice.
WT and mdx mice were studied under normal and hypoxia conditions. Hypoxia reduced resting membrane potential and increased intracellular calcium concentration in VSMC. Comparatively to WT mice, mdx animals were characterized by decreased resting membrane potential and enhanced calcium concentration, both at normoxia and hypoxia. Reduction of extracellular calcium concentration and application of ion channels inhibitors nifedipin and SAR7334 decreased hypoxia-induced calcium overload. The study demonstrates that dystrophin mutation increases the hypoxia-stimulated calcium overload of VSMC, which is due to the activity of L-type Ca+2and TRPC ion channels. This is an interesting study providing useful information.
Minor critique: Figure 1 lacks p values.
Author Response
Response to Reviewer’s Comments
The authors thank the reviewers for their constructive comments, which have significantly strengthened our manuscript. As indicated below, we addressed all specific comments provided by the reviewers and made the necessary changes accordingly to their indications.
Review 1
Minor critique: Figure 1 lacks p values.
We apologize for the error. We have added the p values in Figure 1
Reviewer 2 Report
This manuscript comes from a team who showed as early as 1987 from Caracas, Venezuela that in dystrophic skeletal muscle [Ca2+]i is impaired - one year before Turner et al. and Gillis et al. who received a lot of credit for their “initial” discovery. However, the finding of the team from Caracas remained mostly unnoticed because it was published in Acta científica venezolana, which is not easy to find because it was not indexed in Medline at that time.
After a series of impressive publications, now from Miami, Florida instead of Caracas, they showed in 2020 (Ref. 14) that aortic smooth muscle cells prepared from the most commonly used animal model of DMD, the mdx mouse have, an elevated [Ca2+]I even under basal conditions and as is well-known from other cellular preparations. In the current manuscript, this is further elaborated: Both, normal and mdx VSMC cells were exposed to experimental hypoxia causing a further elevation [Ca2+]i ; this elevation can be inhibited when extracellular calcium levels are reduced to about 1 mM. Partial inhibition can also be achieved by (too) high concentrations of nifedipine or amlodipine or by an experimental compound thought to block some of the TRP channels.
Comments
1. The text of the manuscript is well-written; methods are clearly described, and details are given.
2. I find the last paragraph of the Introduction somewhat tendentious. The stated hypothesis and the postulate are not novel as the link hypoxia – calcium has been shown before in numerous cell types. A more modest formulation is indicated.
3. The cited literature is correct but most of it is 20-30 years old. This is unusual and perhaps original showing the long-time involvement of the senior author in cellular and functional muscular dystrophy research. Although the citations, besides a few exceptions, are correct, there is ample information in the more recent literature.
An example is the missed review from 2016 by Allen, Whitehead & Froehner (doi:10.1152/physrev.00007.2015) that would cover all references on this topic by David Allen. The authors have to streamline the list of references by reducing those that have been outdated or perhaps found to be erroneous and by adding more up-to-date literature.
4. Results: The results shown in Figures 1-3 from the acute manipulations are of no surprise as very similar results have been obtained in many labs with skeletal or cardiac cells. However, what is quite surprising and very welcome is the magnitude of the signals and the large differences between control and mdx cultured VSMCs when compared to skeletal or cardiac cells. These differences between control and mdx VSMCs are up to 10-fold in magnitude, whereas in other cells, they are 2 to 3-fold. – This must be pointed out. Either the protocols have been very well designed or VSMCs are inherently more sensitive to extracellular calcium levels or experimental hypoxia. I am sure the authors have noted this encouraging finding. The manuscript would greatly increase in attractivity if this would be mentioned and commented in the Discussion.
5. VSMCs cultured for 8-10 days in 5% CO2 were used for the experiments and from the results it is concluded that the above-mentioned differences in signals control/mdx are due to the presence or absence of dystrophin. However, these levels or the levels of other candidate proteins have not been given. This requires some numbers and needs to be stated, either experimentally or by citation.
6. The concentration of Ca2+ in the low extracellular Ca2+ concentration experiments is not given.
7. Nifedipine was used at only one high concentration. An interpretation of the result is difficult as nM concentrations of this L-channel blocker are sufficient to inhibit calcium entry; see, e.g. results in ref. 44 where 1-10 nM of nifedipine is effective on downstream function. I appreciate that a note on limitation of the interpretation of results with SAR7334, presumed to be a TRPC3 and TRPC6 blocker is made in the Discussion as a similar problem exists.
8. In lines 235 & 236 in the Discussion, the sentence “Taken together…” is true. However, the same sentence - in various modulations - has been in hundreds of publications. A more modest wording would be of benefit. In agreement with earlier findings of skeletal and cardiac muscle ….
Minor comments
The uploaded pdf file shows them in highlights.
Interesting word creation: dyshomeostasis. What is wrong with dysregulation?
Note: In the publication in Front Physiol 2022: "Chronic Elevation of Skeletal Muscle [Ca2+]i Impairs Glucose Uptake" the last words of the abstract are “...expression of GLUT4 and its subcellular fractionation." it should be “...subcellular location”. Please inform the editors of Front Physiol.

Author Response
Response to Reviewer’s Comments
The authors thank the reviewers for their constructive comments, which have significantly strengthened our manuscript. As indicated below, we addressed all specific comments provided by the reviewers and made the necessary changes accordingly to their indications.
Review 2
1.- We wish to express our gratitude to Reviewer #2 for its encouraging and appreciative comment regarding our publication on [Ca2+] in human muscle fibers of dystrophic patients. Unfortunately, it did not have the desired impact, due, in part, to the factors pointed out by the reviewer. Thank you again.
- I find the last paragraph of the Introduction somewhat tendentious. The stated hypothesis and the postulate are not novel as the link hypoxia – calcium has been shown before in numerous cell types. A more modest formulation is indicated.
We have modified the last paragraph of the Introduction according to the suggestion of the reviewer and now read: 'In the study, we have tested the effects of acute hypoxic stress on [Ca2+]i in WT and mdx VSMCs. We found that mdx VSMCs are more susceptible to hypoxic stress than WT. Susceptibility appears to depend on the anomalous increase in [Ca2+]I, and subsequent reduction in cell viability of mdx VSMCs compared to WT VSMCs. Furthermore, Ca2+ influx can play an essential role as a contributor to intracellular Ca2+ overload observed during acute hypoxia.
- The cited literature is correct but most of it is 20-30 years old. This is unusual and perhaps original showing the long-time involvement of the senior author in cellular and functional muscular dystrophy research. Although the citations, besides a few exceptions, are correct, there is ample information in the more recent literature. An example is the missed review from 2016 by Allen, Whitehead & Froehner (doi:10.1152/physrev.00007.2015) that would cover all references on this topic by David Allen. The authors have to streamline the list of references by reducing those that have been outdated or perhaps found to be erroneous and by adding more up-to-date literature.
In the revised manuscript, we have included the review by Allen, Whitehead & Froehner (2015), and the total number of references was streamlined from 57 to 43.
- Results: The results shown in Figures 1-3 from the acute manipulations are of no surprise as very similar results have been obtained in many labs with skeletal or cardiac cells. However, what is quite surprising and very welcome is the magnitude of the signals and the large differences between control and mdx cultured VSMCs when compared to skeletal or cardiac cells. These differences between control and mdx VSMCs are up to 10-fold in magnitude, whereas in other cells, they are 2 to 3-fold. – This must be pointed out. Either the protocols have been very well designed or VSMCs are inherently more sensitive to extracellular calcium levels or experimental hypoxia. I am sure the authors have noted this encouraging finding. The manuscript would greatly increase in attractivity if this would be mentioned and commented in the Discussion.
[Ca2+]i in WT was 123±3 nM versus 285±31 nM in mdx VSMCs (2.3-fold) which was consistent with our previous report (MID: 32153426). Furthermore, the magnitude of the increase in [Ca2+]i is similar to those observed in striated muscle cells from DMD patients and striated muscle and neurons from mdx mice (PMID: 25181488; PMID: 28623080; PMID: 27975174; PMID: 3506369). The 7.9 and 10.3 -fold increase in [Ca2+]i occurred when the WT and mdx VSMCs were subjected to hypoxia, respectively.
- VSMCs cultured for 8-10 days in 5% CO2 were used for the experiments and from the results it is concluded that the above-mentioned differences in signals control/mdx are due to the presence or absence of dystrophin. However, these levels or the levels of other candidate proteins have not been given. This requires some numbers and needs to be stated, either experimentally or by citation.
The WT and mdx VSMC phenotype was confirmed by demonstrating the expression of muscle myofilament proteins (α-smooth muscle actin and β-tropomyosin) using western blotting (PMID: 22222531). Now it reads 'The VSMC phenotype was confirmed by validating the expression of muscle myofilament proteins (α-smooth muscle actin and β-tropomyosin) using western blotting [14]'.
- The concentration of Ca2+ in the low extracellular Ca2+ concentration experiments is not given.
In the revised manuscript, we have included the actual [Ca2+] in the low extracellular Ca2+solution and read: “For the low Ca2+ solution, CaCl2 was omitted and instead 2 mM MgCl2 and 1 mM ethyleneglycoltetraacetic acid (EGTA) were added (35±8 nM ionic [Ca2+] determined with Ca2+selective microelectrodes).
- Nifedipine was used at only one high concentration. An interpretation of the result is difficult as nM concentrations of this L-channel blocker are sufficient to inhibit calcium entry; see, e.g. results in ref. 44 where 1-10 nM of nifedipine is effective on downstream function. I appreciate that a note on limitation of the interpretation of results with SAR7334, presumed to be a TRPC3 and TRPC6 blocker is made in the Discussion as a similar problem exists.
The purpose of the nifedipine experiments was to block the Ca2+ influx mediated by the L-type Ca2+ channel. Namely, dissect this pathway from the other that exists in the sarcolemma pharmacologically. We found that in VSMCs, nifedipine (2 - 8 μM) reduced the amplitude of Ca2+ currents in a dose-dependent manner but did not completely inhibit it. Only at high concentrations (10 µM) were Ca2+ currents fully inhibited. Because Ca2+ currents from VSMCs have small amplitudes, we used 10 mM Ba2+ as the current carrier ion. Unfortunately, the use of 10 mM may modify the effect of nifedipine on the amplitude of the Ca2+ current since the composition of extracellular divalent cations can alter the inhibition of the Ca2+ current by nifedipine. It should be noted that nifedipine at such a concentration could interact with other ion channels such as TRIP-A1 (PMID: 18971630). We have addressed this issue in the limitations section.
Regarding SAR7334, 1 µM was chosen because this concentration blocks TRPC3 and 6-channel conductance (PMID: 25847402).
- In lines 235 & 236 in the Discussion, the sentence “Taken together…” is true. However, the same sentence - in various modulations - has been in hundreds of publications. A more modest wording would be of benefit. In agreement with earlier findings of skeletal and cardiac muscle ….
We agree with the reviewers' comments and the sentence 'Taken together' has been removed from the body of the manuscript and now reads 'Thus, the weight of evidence supports the notion that [Ca2+]i plays a crucial role in the pathogenesis of DMD, and preventing its overload reduces muscle injury and minimizes hypoxia-induced cell damage.'.
Minor comments
- The uploaded pdf file shows them in highlights.
We will do it on the revised manuscript
- Interesting word creation: dyshomeostasis. What is wrong with dysregulation?
We appreciate comments on the creative word 'dyshomeostasis'. Although dyshomeostasis is defined as an imbalance of a homeostasis system, the work was removed from the text of the manuscript.
Note: In the publication in Front Physiol 2022: "Chronic Elevation of Skeletal Muscle [Ca2+]i Impairs Glucose Uptake" the last words of the abstract are “...expression of GLUT4 and its subcellular fractionation." it should be “...subcellular location”. Please inform the editors of Front Physiol.
We welcome this comment. We used it because in the scientific literature, “subcellular fractionation” has been used as a method to separate the intracellular membrane and the plasma membrane.
Round 2
Reviewer 2 Report
Authors have answered all queries and modified the manuscript, which has been much improved. Even a figure has been added.
I have only a few minor linguistical comments:
Line 70 – remove “that”
Lines 167-169 – make 2 sentences as it could be misunderstood that the electrode containing the sensor was heated with it to 150 degrees.
Lines 249 and 255 – replace “low” with the 35±8 nM value mentioned in Methods to facilitate reading.
Lines 280 & 281 – Ca2+
Lines 301 & 28 - replace “increases” by “increased” and check manuscript for constant case, i.e. past when referring to a past observation
Author Response
Point by point response:
Authors have answered all queries and modified the manuscript, which has been much improved. Even a figure has been added.
I have only a few minor linguistical comments:
We thank the reviewer for this thorough review and comments
Line 70 – remove “that”-
This has been done
Lines 167-169 – make 2 sentences as it could be misunderstood that the electrode containing the sensor was heated with it to 150 degrees.
The two new sentences now read:
The Ca2+ microelectrode barrel was made hydrophobic by silanization with dimethyldichlorosilane vapor (Sigma-Aldrich, MO, USA) and baked at 150o C for 1 h. Subsequently, the salinized tip of the microelectrode was filled with a liquid ion sensor ETH 129 (Fluka-Sigma-Aldrich, MO, USA), and the rest of the barrel was filled with pCa7.
Lines 249 and 255 – replace “low” with the 35±8 nM value mentioned in Methods to facilitate reading.
This has been corrected
Lines 280 & 281 – Ca2+
This has been corrected
Lines 301 & 28 - replace “increases” by “increased” and check manuscript for constant case, i.e. past when referring to a past observation
This has been corrected, we have also used two software analysis programs for grammar and punctuation checks